# Flavonoid Synthesis-Related Genes Determine the Color of Flower Petals in *Brassica napus* L.

**DOI:** 10.3390/ijms24076472

**Published:** 2023-03-30

**Authors:** Shijun Li, Xi Li, Xiaodan Wang, Tao Chang, Zechuan Peng, Chunyun Guan, Mei Guan

**Affiliations:** National Oil Crops Improvement Center in Hunan, Department of Agronomy, Hunan Agricultural University, Changsha 410128, China

**Keywords:** *Brassica napus*, petal color, transcriptome sequencing, ultra-performance liquid chromatography-tandem mass spectrometry (UPLC-MS/MS)

## Abstract

The color of rapeseed (*Brassica napus* L.) petal is usually yellow but can be milky-white to orange or pink. Thus, the petal color is a popular target in rapeseed breeding programs. In his study, metabolites and RNA were extracted from the yellow (Y), yellow/purple (YP), light purple (LP), and purple (P) rapeseed petals. Ultra-performance liquid chromatography-tandem mass spectrometry (UPLC-MS/MS), RNA-Seq, and quantitative real-time (qRT-PCR) analyses were performed to analyze the expression correlation of differential metabolites and differential genes. A total of 223 metabolites were identified in the petals of the three purple and yellow rapeseed varieties by UPLC-MS/MS. A total of 20511 differentially expressed genes (DEGs) between P, LP, YP, versus Y plant petals were detected. This study focused on the co-regulation of 4898 differential genes in the three comparison groups. Kyoto Encyclopedia of Genes and Genomes (KEGG) functional annotation and quantitative RT-PCR analysis showed that the expression of *BnaA10g23330D* (*BnF3'H*) affects the synthesis of downstream peonidin and delphinidin and is a key gene regulating the purple color of petals in *B. napus.* L. The gene may play a key role in regulating rapeseed flower color; however, further studies are needed to verify this. These results deepen our understanding of the molecular mechanisms underlying petal color and provide the theoretical and practical basis for flower breeding targeting petal color.

## 1. Introduction

Rapeseed (*Brassica napus* L.) is an important oil crop in China and is the most produced oil globally [1,2]. In addition to the common oil, rapeseed is used as food, fertilizer, ornamental plant, and a target for agricultural tourism. The traditional rapeseed flowers are mostly yellow or white. However, in recent years, breeders have obtained different colors, such as orange, red, and purple, through artificial selection, distant hybridization, mutagenic breeding, and genetic engineering.

The different floral colors are regulated by one or several pigments, such as carotenoids, flavonoids, and others. Carotenoids and flavonoids are widely distributed plant pigments responsible for the yellow or orange flower colors. Flavonoids are major secondary metabolites derived from the phenylpropanoid pathway and can be further classified into six main groups—flavones, flavonols, flavanones, flavanols, anthocyanidins, and proanthocyanidins [3]. In recent years, increasing evidence has demonstrated crosstalk between flavonoid and carotenoid pathways, producing metabolites with novel biological effects. In rapeseed flowers, the petal color varies and could be regulated through natural pollination, artificial mutagenesis, and gene transfer. Novel floral colors derived from natural mutation are mainly orange-red, golden-yellow, or milky white [4]. Moreover, genetic transformation has been used to obtain new floral colors, such as purple, pink, and red [5].

Tian Lushen et al. revealed that genes such as *PSY*, *PDS*, and *ZDS* modulate carotenoid synthesis, and *NCED3* and *CCD1*, which participate in carotenoid degradation, regulate the flower color of rapeseed [6]. Zhang et al. used the positional cloning method to identify a carotenoid cleavage double oxygenase gene (*BnaC3.CCD4)* that regulates the floral color of *Brassica napus* [7]. Using near-isogenic lines, Zhang et al. found that *BjPC1* and *Bjpc2* genes regulate the white pigmentation trait in rapeseed mustard and the secretion of carotenoids [8]. Moreover, Jia et al. showed that *BnNCED4b* encodes a protein involved in carotenoid degradation, and this protein is overexpressed in *Brassica napus* L. [9]. These findings demonstrate that the development of white-flowered rape is closely related to the metabolism of carotenoids. For instance, Lee et al. revealed that *BrCRTISO1*, a carotenoid isomerase gene, regulates the pigmentation of rape petals, an orange-leaved cabbage [10]. Wu et al. determined the carotenoid content in the petals of four different flower-colored petals (golden, yellow, light yellow, and white) in kale rape and identified five genes (*BnPSY*, *BnLYCB*, *BnLYCB*, *BnLYCE*, *BnZEP*, and *BnNCED*), which are closely related to carotenoid synthesis or degradation [11]. Besides carotenoids, there is evidence that genes in the flavonoid pathway may also play a role in rape floral color. For example, Xu et al. showed that the *BnF3′H* gene associated with flavonoid synthesis in rapeseed is mainly expressed in petals [12]. In addition, Qiao Cailin et al. demonstrated that the genes that regulate the orange color in *Brassica napus* L. are highly expressed in the anthocyanin synthesis pathway [13].

The regulation of petal color in rapeseed is a complex process, yet only a few genes that participate in this process have been identified. We previously identified a natural purple-flowered mutant in an inbred rapeseed line. Rapeseed is an ornamental crop whose petal color has remained stable through several self-breeding generations (Figure 1). To identify the key genes and mutations regulating the purple floral traits, we sequenced the DNA of pure inbred lines with yellow/purple, light purple, purple, and yellow flowers (YP, LP, P, and Y, respectively). Differentially expressed genes (DEGs) among the rapeseed species were identified. Gene ontology (GO) and the Kyoto Encyclopedia of Genes and Genomes (KEGG) analyses of the DEGs were performed to identify the biological processes and pathways regulated by these genes. Our analyses identified genes that potentially play vital roles in determining petal color. The results of this study provide insight into the molecular basis of petal coloration in *B. napus* and lay a foundation for breeding purple-flowered *B. napus* varieties with high ornamental value.

## 2. Results

### 2.1. The Anthocyanin and Carotenoid Contents in the Petals of the Four Brassica napus L. Cultivars

The contents of anthocyanins in LP, YP, and P petals were significantly higher than that in Y petals. The contents of carotenoids in the three purple petals were significantly lower than that in Y petals. These results indicated that anthocyanin contents were higher in deep purple petals, whereas deep yellow petals had higher carotenoid contents (Figure 2).

### 2.2. Widely Targeted Metabolomic Analysis of the Four Brassica napus L. Cultivars

In this study, the grain colors of the four plant materials Y, YP, LP, and P were yellow, yellow/purple, light purple, and purple, respectively. Metabolites in the samples were detected by UPLC-MS/MS, and qualitatively and quantitatively analyzed by MS. The total ion current (TIC) for the QC sample mixture and multi-peak diagram of MRM metabolite detection are shown in Appendix A. The MRM metabolite detection multi-peak diagram shows all substances detected in the samples. Of the 223 metabolites, the larger proportion was 118 flavonols, 51 flavones, 21 flavanones, and 15 anthocyanidins (Appendix A). Analysis of the TIC graphs (Figure 3A), plotted for MS detection and the various QC samples, showed that the TIC metabolite detection curves had a high degree of overlap. Hence, retention times and peak intensities were equal among samples, and MS occurred at different times. Spot detection of the same samples demonstrated stable signals. Thus, metabolite extraction and detection were reliable. Pearson’s correlation coefficient r was used to evaluate the biological repetition of samples within each group. As *r* 2 approached unity, the stronger the correlation between the two replicate samples (Figure 3B). A cluster heat map analysis was performed on all metabolites. The sample population cluster heat map showed that all three biological replicates of each species were clustered together (Appendix A). The PCA revealed the degree of metabolic variation between groups and among samples within the same group. Y, YP, LP, P, and their combinations within the distribution of apparent dispersion showed that the metabolite levels differed among samples. P, LP, and YP were distinctively separated from Y between PC1 and PC2 (Figure 3C). Therefore, the materials were sufficiently reproducible, suitable for use in the subsequent qualitative and quantitative analyses, and adequate to ensure the repeatability and reliability of the metabolomic data.

### 2.3. Metabolic Differences among the Four Petal Colours of Brassica napus L.

Differential metabolism showed 3, 21, and 16 unique metabolites in Y vs. LP, Y vs. P, and Y vs. YP, respectively. There were 103 metabolites that significantly differed in Y vs. LP. Of these, the expression of 92 was upregulated and that of 11 was downregulated. There were 130 significantly different metabolites in Y vs. P. Of these, the expression of 115 was upregulated and that of 15 was downregulated. There were 108 significantly different metabolites in Y vs. YP. Of these, the expression of 89 was upregulated and that of 19 was downregulated (Figure 4). 

A PCA was performed on the samples, and significant separation of the first principal component was observed among groups (Appendix A). There was significant metabolite accumulation among the samples, and primarily, the contents of flavonoids had changed (Figure 5A). The metabolites showed contrasting expression patterns among the differently colored rapeseed petals (Figure 5B). Flavonoid content was higher in purple-colored rapeseed than in yellow-colored rapeseed (Appendix A). The *K* means analysis grouped metabolites with the same changing trend, revealing that the levels of most flavonoids were generally higher in the three purple rapeseeds than in the yellow rapeseeds. The KEGG analysis annotated and classified the significantly different metabolites according to the pathways to which they belonged. Most of the significantly different metabolites were categorized into the metabolic pathway, followed by the specialized metabolite pathway. 130 differential metabolites were annotated into the flavonoid and flavonol biosynthesis pathway, anthocyanin biosynthesis pathway, and flavonoid biosynthesis pathway (Figure 5D). Some representative substances include peonidin-3-O-glucoside, delphinidin-3-O-glucoside, and kaempferol-7-O-rhamnoside. The genes regulating these metabolites include *F3′H*, *FLS,* and *UFGT,* among others.

### 2.4. Transcriptome and Differential Gene Expression Analyses

For the transcriptome sequencing analysis of the twelve rapeseed samples, the sample correlation heat map (Appendix A) and PCA diagram (Appendix A) demonstrated suitable biological reproducibility within each rapeseed line and distinct differences among rapeseed lines. A total of 570,739,860 clean reads were obtained from three replicas of purple (P), light purple (LP), yellow/purple (YP), and yellow (Y) petals of rapeseed flower (12 samples in total) (Figure 1A,B). The error rate for all samples was less than 0.03%. The proportion of Q20 clean reads (correct base recognition rate higher than 99%) and Q30 (correct base recognition rate higher than 99.9%) was higher than 97% and 92%, respectively. For LP, 90.45% of the clean reads were mapped onto the *B. napus* reference genome. The matching rates were 90.89% for P; 93.59% for Y, and 90.62% for YP (Appendix A). 

A total of 10,691 DEGs were identified between yellow and light purple petals (Y vs. LP) (4013 and 6678 upregulated and downregulated, respectively). Additionally, 10,980 DEGs were identified between yellow and yellow/purple petals (Y vs. YP) (4854 and 6126 upregulated and downregulated, respectively). Finally, 10,660 DEGs were identified between yellow and purple petals (Y vs. P) (4087 and 6573 upregulated and downregulated, respectively) (Figure 6A). The Venn diagram identified 4898 DEGs among the three groups (Figure 6B). 

### 2.5. GO and KEGG Pathway Analysis of the DEGs

GO analysis identified the molecular function, cellular components, and biological processes regulated by the DEGs (Figure 7). A total of 4641 (15.70%) DEGs between Y and LP, 4758 (15.76%) DEGs between Y and YP, and 4686 (15.66%) DEGs between Y and P, regulated several metabolic processes. Therefore, different degrees of purple petals may not greatly affect the number of DEGs involved in the metabolic process in *Brassica napus*, and the proportion of DEGs involved in metabolism is comparable to that of the total DEGs among the three petal subtypes. KEGG pathway analysis showed that nine DEGs were annotated into flavonoid and anthocyanin biosynthesis pathways (Appendix A). The expression of *F3′H* in the three purple petals was significantly higher than that of the yellow petals. The expressions of *FLS*, *F3H*, *VSR*, *HCT,* and *UFGT* genes also differed between the purple and yellow petals.

### 2.6. Combined Transcriptomic and Metabolomic Analyses

Metabolome and transcriptome data were integrated to clarify the differences in petal color formation among the *B. napus* varieties. The KEGG analysis showed that the differential metabolites and DEGs related to color formation were present in flavonoid biosynthesis. A canonical correlation analysis of flavonoid compound biosynthesis showed that the DEGs and differential metabolites were highly correlated in all three groups (Figure 8A). An integrated O2PLS analysis of the transcriptome and metabolome datasets is shown in Figure 8B. Joint KEGG analysis of transcriptome and metabolome pathways showed that upregulation of the *F3′H* gene affected the increase of downstream anthocyanin content (Figure 9). As a consequence of the transcription and metabolic group mutual influence, alterations in transcriptome data variables strongly affected the metabolomics. Integrated metabolomic and transcriptomic analyses of the flavonoid biosynthesis pathway involved in the formation of *B. napus* petal colors (Figure 10) revealed the differences in different genes and metabolites under the flavonoid biosynthesis pathway. These discrepancies resulted in different levels of peonidin, cyanidin, and delphinidin, and by extension, different anthocyanin biosynthesis pathways among *B. napus* petal types.

### 2.7. qRT-PCR Analysis of DEGs in the Flavonoid Metabolic Pathway

To verify the differential expression of the genes identified by transcriptome analysis, we used qRT-PCR to analyze the relative expression levels of nine selected genes involved in flavonoid metabolism (*HCT*, *VSR7*, *F3H*, *F3′H*, *FLS,* and *UFGT*). The FPKM curve in *BnaA08g27660D* did not match the qRT-PCR, and the expression of *BnaC09g47360D* did not differ significantly between Y and YP (Appendix A). The relative expression of other genes was basically consistent with the transcriptome results, among which, the *F3′H* gene was the key gene for regulating purple flower traits. The relative expression of the *BnaA10g23330D* gene in LP, YP, and P was significantly higher than that in Y. Transcriptome results showed that the relative expression of gene *BnaA10g23330D* in Y, LP, YP, and P was 1, 11.93, 9.38, and 8.17, respectively (Appendix A). qRT-PCR and RNA-seq results were generally consistent (Figure 11).

## 3. Materials and Methods

### 3.1. Plant Materials

The yellow rapeseeds were provided by Hunan Agricultural University, the rest of rapeseeds were provided by Professor Fu Donghui of Jiangxi Agricultural University. YP, LP, and P inbred lines were selected from a group with the same genetic background. The yellow/purple petal (YP), light purple (LP), purple (P), and yellow (Y) (variety XY708) homozygous inbred rapeseed lines (*Brassica napus*) were planted in the experimental field of the Hunan Agriculture University, Yunyuan. In March 2022, newly opened petals were sampled from each of the four lines based on phenotypic appearance (Figure 1A). Three replicas were sampled from each group (totaling 12 samples). All samples were measured in triplicates for three independent biological replicates and stored at −80 °C till analysis.

### 3.2. Metabolomic Analysis

#### 3.2.1. Determination of Carotenoid and Anthocyanin Contents

Anthocyanins were extracted using aqueous methanol containing 0.1% hydrochloric acid extraction solution. The supernatants were stored at 4 °C for 12 h in the dark. The supernatant was filtered, and its absorbance was measured at 530 and 657 nm by an ultraviolet spectrophotometer (Hoefer Vision, SP-2001). The total anthocyanin content was calculated as QAnthocyanins = [A530 − (0.25 × A657)]/M, where QAnthocyanins is the amount of anthocyanins, A530 and A657 are the absorbances at the indicated wavelengths, and M is the weight of freeze-dried powder sample used for extraction [14]. All analyses were performed in triplicates [15]. Carotenoids were extracted using the Plant carotenoid content Assay Kit (ZCiBio, Shanghai, China). Its absorbance was measured at 440 nm by an ultraviolet spectrophotometer (Hoefer Vision, SP-2001). The total anthocyanin content was calculated as QCarotenoids = 0.04 × A440 × F ÷ W, where carotenoids are the amount of carotenoids, A440 is the absorbance at the indicated wavelength, W is the fresh weight of the plant material used for extraction, and F is the dilution factor. All samples were measured in triplicates from three independent biological replicates. Data are means ± SD obtained from three biological replicates. Significant differences between different samples were determined at *p* < 0.05, according to Tukey’s test.

#### 3.2.2. Metabolite Extraction and Ultra Performance Liquid Chromatography-MS/MS Conditions

Biological samples were freeze-dried with a vacuum freeze-dryer (Scientz-100F). The freeze-dried samples were crushed using a mixer mill (MM 400, Retsch) with a zirconia bead for 1.5 min at 30 Hz. Lyophilized powder (50 mg) was dissolved with a 1.2 mL 70% methanol solution, vortexed for 30 s every 30 min 6 times, and then placed in a refrigerator at 4 °C overnight. Following centrifugation at 12,000 rpm for 3 min, the extracts were filtrated (SCAA-104, 0.22 μm pore size; ANPEL, Shanghai, China, http://www.anpel.com.cn/, accessed on 18 April 2022) before UPLC-MS/MS analysis.

The concentration of specific metabolites in the sample extracts was analyzed using the UPLC-MS/MS system (Shimadzu Nexera X2, Applied Biosystems 4500 QTRAP) and the column ACQUITY UPLC HSS T3 C18 (pore size 1.8 µm, length 2.1 × 100 mm). The solvent system contained a mobile phase A (0.1% acetic acid in water) and B (0.1% acetic acid in acetonitrile). The gradient program was 95:5 *V*/*V* at 0 min, 5:95 *V*/*V* at 9.0 min, 5:95 *V*/*V* at 10.0 min, 95:5 *V*/*V* at 11.1 min, and 95:5 *V*/*V* at 14 min. The flow rate was 0.35 mL/min at 40 °C, and the injection volume was 4 µL. Data were acquired using multiple reaction monitoring (MRM) with a triple quadrupole tandem mass spectrometer and processed using Analyst software (version 1.6.3, Sciex) [16]. All samples were measured in triplicates for three independent biological replicates. 

#### 3.2.3. Qualitative and Quantitative Metabolite Analyses

Qualitative metabolite analysis was performed using the in-house Metware database (Metware Biotechnology Co., Ltd., Wuhan, China, accessed on 21 May 2022) according to secondary spectral information. Isotopic signals, repeated signals containing K^+^, Na^+^, and NH^4+^ ions, and repeated signals of other high-MW debris ions were removed during the analysis. Metabolites were quantified by multiple reaction monitoring (MRM) analysis. During the instrumental analysis, a quality control (QC) sample was inserted after every tenth test and sample. The repeatability of the total ion flow detection method was determined by testing the spectra of various QC samples. Principal component (PCA) and orthogonal partial least squares discriminant analyses (OPLS-DA) were conducted on all metabolites to identify putative biomarkers. Metabolites with significantly different metabolism were selected as biomarkers based on Fold Change ≥ 2 and Fold Change ≤ 0.5. All samples were measured in triplicates from three independent biological replicates.

### 3.3. Transcriptome Sequencing and Data Analysis

#### 3.3.1. RNA Extraction and RNA-Seq

Total RNA was extracted from the samples using the RNA prep Pure Plant kit (DP441, Tiangen, China) and sequenced by Metware Biotechnology Co., Ltd. using the Illumina RNA-Seq platform (Wuhan, China). The RNA quality was assessed by a Nano Photometer (IMPLEN, CA, USA), Qubit 2.0 Fluorometer (Life Technologies, CA, USA), and Agilent Bioanalyzer 2100 system (Agilent Technologies, CA, USA). The poly (A) mRNAs were hybridized with an oligo (dT). The mRNAs were randomly fragmented and reverse transcribed to cDNA using the M-MuLV reverse transcriptase system. The RNA strand was then degraded by RNase H before the synthesis of the second cDNA strand using DNA polymerase. The double-stranded cDNAs were then ligated to sequencing adapters. The cDNAs (~200 bp) were screened using AMPure XP beads. After amplification and purification, cDNA libraries were obtained and sequenced using the Illumina Novaseq6000 system as described by Chen et al. [17]. The effective concentration of the library was greater than 2 nM. PCR was conducted using EasyScript^®^ One-Step gDNA Removal and cDNA^®^ Synthesis SuperMix (Transgen, Changsha, China) in a typical 25 μL PCR mixture that included 2 μL of template cDNA, 12.5 μL of 2 ×TransTaq ^®^ HiFi PCR SuperMixⅡ, 8.5 μL ddH_2_O and 1 μL of each PCR primer (10 μM). Cycling conditions were 94 °C for 30 min, followed by 37 cycles of 94 °C for 30 s, 60 °C for 30 s, and 72 °C for 1 kb/min. The final extension was at 72 °C for 7 min. All samples were measured in triplicates from three independent biological replicates.

#### 3.3.2. Transcriptome Sequencing

The raw sequence data were transformed by CASAVA base recognition. To obtain high-quality data, the sequence adapters were cut, and low-quality reads with more than five uncertain bases or with over 50% Qphred ≤ 20 bases were removed using fastp. The GC content of clean reads was calculated. The base quality was evaluated with FastQC based on the Q20 and Q30 values. The clean reads were mapped to the hickory reference genome using HISAT2 under default parameters. Gene expression levels were determined using the RPKM (reads per kb per million reads) method [14]. All samples were measured in triplicates from three independent biological replicates.

#### 3.3.3. qPCR Validation of the DEGs

The expression of DEGs related to flavonoid metabolism, including flavonoid 3-hydroxylase (*BnF3H*), flavonoid 3′-hydroxylase (*BnF3ʹH*), flavonol synthase (*BnFLS*), hydroxycinnamoyl-CoA shikimate/quinate hydroxycinnamoyl transferase (*BnHCT*), flavonoid 3-O-glycosyltransferase (*BnUFGT*), and vacuolar sorting receptor (*BnVSR*) genes were validated using qPCR. RNA was extracted from the petals of the rapeseed lines as described in Section 3.2, and cDNA was synthesized from 2 µg of each RNA sample using the TransScript One-Step gDNA Removal and cDNA Synthesis SuperMix (Transgen, Changsha, China). The PCR primers were designed using the NCBI platform (Appendix A) and synthesized by Qingke Biotechnology Co., Ltd., (Changsha, China). The qPCR was conducted on the CFX96TM Real-Time PCR System (BIO-RAD, USA) using TransScript Tip Green qRT-PCR SuperMix (Transgen, Changsha, China) in a typical 10 μL PCR mixture that included 1 μL of template cDNA, 10 μL of 2 × TransStart^®^ Tip Green qPCR SuperMix, 0.4 μL of Passion Reference Dye (50 ×), 7.8 μL ddH2O, and 0.4 μL of each PCR primer (10 μM). Cycling conditions were 94 °C for 30 s, followed by 40 cycles of 94 °C for 5 s (denaturation), and 60 °C for 30 s (annealing and extension). Relative expression of genes was analyzed by the 2^−ΔΔCT^ method [18] using *BnACTIN7* (AF111812; forward primer: 5′-GGTTGGGATGGACCAGAAGG-3′, reverse primer: 5′-TCAGGAGCAATACGGAGC-3′) as an internal control [19]. Data were analyzed using the SPSS software v. 20. Differences in Y among groups were analyzed using the t-test. Graphs were plotted using GraphPad Prism8. All samples were measured in triplicates from three independent biological replicates.

#### 3.3.4. Transcriptome Data Analysis

A differential expression analysis was conducted on the various sample groups. FPKM values were used to calculate the expression level of genes. DESeq2 v1.22.1/edgeR v3.24.3 was used to analyze the differential expression between the two groups, and the *p* value was corrected using the Benjamini & Hochberg method. The corrected *p* value and |log2foldchange| were used as the threshold for significant differential expression. |log2FoldChange > 1 and *p* ≤ 0.05 were determined as differentially expressed genes (DEGs). Venn diagram of DEGs, PCA, GO, and KEGG was also conducted for further analysis. All samples were measured in triplicates from three independent biological replicates.

### 3.4. Integrated Metabolomic and Transcriptomic Analyses

The transcriptome and metabolome data were normalized and statistically analyzed to establish the relationships between the genes and the metabolites implicated in wheat grain color development. PCA, Kyoto Encyclopedia of Genes and Genomes (KEGG) pathway, correlation, two-way orthogonal partial least squares (O2PLS), and other analyses were also conducted. Interactive comparisons of metabolomic and transcriptomic data identified potential metabolites and their corresponding DEGs at the molecular and biochemical pathway levels. All samples were measured in triplicates from three independent biological replicates.

## 4. Discussion

In the present study, we measured the carotenoid content in three purple rape and yellow rape (Appendix A). The results showed a positive correlation between the yellow pigmentation intensity and the carotenoid content. Liu et al. reported that crosstalk between the carotenoid and flavonoid pathways regulates the color of *Brassica napus* petals [20]. A previous study also showed that the yellow pigmentation of *Brassica napus* flowers is mediated by carotenoid pigments [7]. This implied that the carotenoids regulate the yellow pigmentation trait, consistent with previous findings.

However, carotenoids are not central to the purple flower traits [21]. There is a need for further studies to unravel the potential role of flavonoids on the color of *B. napus* flowers. Flavonoid biosynthetic pathways have been extensively characterized in higher plants. Herein, RNA-Seq technology identified 33 genes that were differentially expressed in different *B. napus* cultivars (Figure 8), indicating that the different colors were regulated by specific genes. In the anthocyanin biosynthetic pathway, trans-cinnamic acid is initially formed through the deamination of phenylalanine by phenylalanine ammonia-lyase (*PAL*). Cinnamoyl-CoA and p-coumaroyl-CoA are then produced in a reaction catalyzed by 4-coumarate-CoA ligase (4CL) and trans-cinnamate 4-monooxygenase (C4H) [22]. Next, p-coumaroyl-CoA is isomerized to flavanone, catalyzed by chalcone synthase (*CHS*) and chalcone isomerase (*CHI*) [23]. Finally, the flavanones are converted to dihydroflavonols, catalyzed by flavanone 3-hydroxylase (*F3H*). According to the transcriptome analysis results, the expression of *C4H* and *CHI* was significantly decreased in purple and light purple-colored rape petals compared to yellow rape petals. Dihydroflavonols are converted into pelargonidin, cyanidin, and delphinidin, involved in anthocyanin biosynthesis, catalyzed by flavonoid 3′, 5′-hydroxylase (*F3′5′H*), flavonoid 3′-monooxygenase (*F3′H*), dihydroflavonol 4-reductase (*DFR*), anthocyanidin synthase (*ANS*), and flavonoid-O-glycosyl-transferase (*UFGT*).

Nishihara et al. induced a flower color change from violet to pale blue or pale violet in *Torenia fournieri* by knocking out the flavanone 3-hydroxylase (*F3H*) gene [24]. However, in *B. napus*, only one successful attempt to alter the flower color via plant transformation has been reported. Transformation of yellow-flowered *B. napus* with *PAP2* (production of anthocyanin pigment 2) from *Orychophragmus violaceus* under the control of a petal-specific promoter promoted the accumulation of anthocyanins in petals, which resulted in the production of red flowers [25]. Takahashi et al. showed that the anthocyanin content of soybean with dark purple flowers was 50% higher than that of purple petals [26], and the anthocyanin content and the development of dark purple flowers were regulated by the *Wd* gene. Liu et al. found that RS392880, an allele of the *F3′H* gene, regulates the purple flower trait of radishes via the anthocyanin biosynthesis pathway [27]. These findings clarify the molecular mechanism underlying petal coloration. Our transcriptomic analysis showed that the expression of *F3*′*H* in the three purple petals was significantly higher than in the yellow petals. Our metabolomic analysis showed that the contents of peonidin-3-O-glucoside, delphinidin-3-O-glucoside, luteolin-7-O-(6′′-caffeoyl) rhamnoside, kaempferol-3-caffeoyl diglucoside, quercetin-4′-O-glucuronide, and kaempferol-7-O-rhamnoside were very high in purple petals compared to the yellow petals. Anthocyanin, a water-soluble natural pigment and a strong antioxidant in humans, is widely present in plants [28]. Over the years, studies have reported the pharmacological properties of anthocyanin [29,30]. Anthocyanins are the glycosylated form of anthocyanidins. We found that the content of anthocyanins and anthocyanidins in the three purple petals was higher than that of yellow petals, but the accumulation of anthocyanins in the three purple petals differed from the accumulation of anthocyanidins. This may be attributed to the limited scope of the detection technology. At present, the detectable anthocyanin substances in the flavonoid metabolism group cannot cover all the anthocyanin categories in nature. The composition and proportion of common co-pigment flavonoids, especially flavonol and flavonoids, impact the color of flowers [31,32]. The importance of flavonoids as auxiliary pigments has been demonstrated in blue flowers of many species, including carnations, coriander, and *Nemophila menziesii* [33,34,35,36]. Our metabolic analysis revealed that although flavonol was the dominant flavonoid, other than anthocyanins, the flavonol content was very low. In addition, the influence of vacuole pH on the blue coloration of *B. napus* flowers cannot be ignored [35]. These findings generally indicate that natural changes in flavonoids, such as downregulation or inactivation of key enzymes, can produce novel and bright colors. The enzymes will cause flux to be reoriented along different branches, thereby forming an appropriate co-pigment composition.

Integrated transcriptomic and metabolomic analysis has become a common tool for assessing the interactions between genes and certain traits. For example, transcriptome and metabolome analyses have revealed how the petal pigmentation trait in rape and soybean is regulated [35]. Considering that the mechanism underlying petal pigmentation in rape had been previously reported [9], we used transcriptomics and metabolomics analyses to unravel this phenomenon. Our results showed that the purple pigmentation of *B. napus* petals is regulated by flavonoids. It is worth noting that the rapes used in the present study were grown in the natural environment to minimize the environmental interference of controlled conditions. Although rape grows in a complex and unstable natural environment in the field, and its physiological changes may be affected by biological and abiotic stresses, we believe that our results reflect the mechanism of rape purple flowers in the natural environment [13]. Our transcriptomic and metabonomic analyses showed that the expression of many flavonoid-related genes and metabolites of rape was significantly upregulated, and the metabolites overexpressed on purple-flowered rape were significantly different from those of yellow-flowered rape.

The main findings of this study were: (i) Upregulation of *BnaC09g30320D* and *BnaC08g08350D* (*UFGT*) promotes the production of anthocyanins in the petals of yellow/purple *B.napus*. (ii) Upregulation of *BnaCnng18670D* (*VSR*) in light purple and purple rape affects butin synthesis. Butin has numerous pharmacological properties, such as anti-inflammatory, antioxidant, endothelium-dependent vasodilation, inhibition of protein hormone, and glutathione reductase activity. It also inhibits the invasion and metastasis of cancer cells by inducing apoptosis and cell cycle arrest. In addition, butin enhances the pharmacological activity of anti-tumor drugs and reduces the drug resistance of cancer cells [37]. (iii) Upregulation of *BnaC08g00430D* (*HCT*) affects the increased phloretin content in yellow/purple rape petals. Phloretin is a dihydrochalcone compound mainly found in apples, pears, strawberries, and other fruits and vegetables. It has several biological functions, including anti-bacterial, anti-viral, tyrosinase inhibition, hypoglycemic, and anti-tumoral properties. Given that phloretin is applied in the food, medical, and cosmetic industries, the yellow/purple-colored rape petals could also be applied for similar uses [38].

Numerous *F3′H* gene mutants have been isolated in many plant species, such as *A thaliana* [39,40], barley [41], and potato [42], based on flower colors, and their functions have been identified. This study analyzed the four most common *Brassica napus* lines with different petal colors. Results revealed that three genes, including *BnaA10g23330D* (*F3′H*), *BnaC09g47360D* (*F3H*), and *BnaC08g33510D* (*FLS*), determine the purple petal phenotype. These findings deepen our understanding of the molecular mechanism underlying the variation of *B. napus* flower color and provide strategies for modifying flower color through genetic transformation [43,44,45]. Moreover, this study provides a reference for the mechanism of purple flower petal formation in *B. napus*, and the metabolites regulating purple flower characters are consistent with previous studies.

## 5. Conclusions

This study combined metabolomic, transcriptomic, and qRT-PCR analyses to explore the mechanisms underlying the differential accumulation of flavonoids in Y, LP, YP, and P petals. The contents of total peonidin and delphinidin in the three purple petals were significantly higher than those in the yellow petals. Transcriptome analysis revealed nine DEGs in the flavonoid metabolism pathway. Among them, *BnF3'H*, a gene involved in anthocyanidin degradation, represents a key candidate gene for the formation of purple petals. The qRT-PCR analysis results were consistent with the RNA-seq results and provided important insights into the molecular mechanisms of the flavonoid metabolic pathway in *B. napus* petals. In addition, the candidate genes identified in this study provide a resource for the development of new rapeseed varieties with different flower colors.

## Figures and Tables

**Figure 1 ijms-24-06472-f001:**
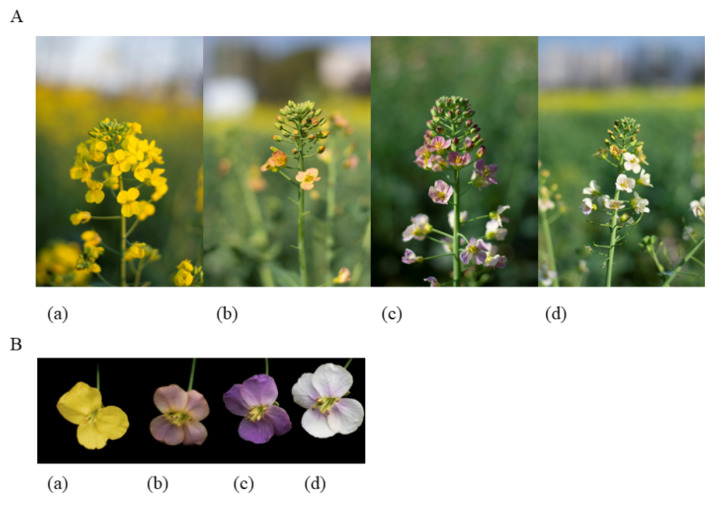
The phenotypes of the four differently-colored flowers of *Brassica napus* (**A**) The wild phenotype (**B**) Rapeseed petals. (**a**) Yellow petals, (**b**) yellow/purple petals, (**c**) purple petals, (**d**) light purple petals.

**Figure 2 ijms-24-06472-f002:**
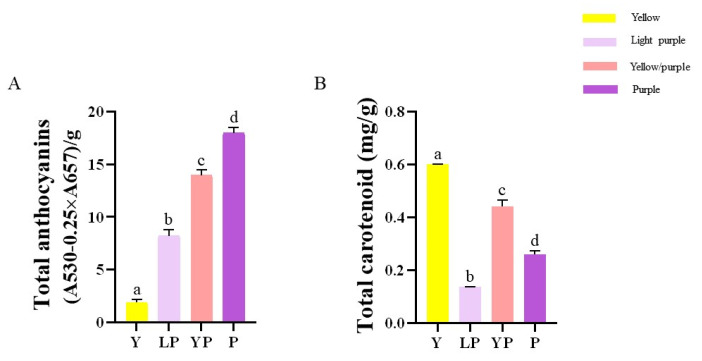
The physiological indicators. (**A**) Anthocyanin extracts in the different petals during the full flowering period. (**B**) Carotenoid extracts in the petals during the full flowering period. Data are means ± SD obtained from three biological replicates. Bars with different letters are significantly different at *p* < 0.05, according to Tukey’s test.

**Figure 3 ijms-24-06472-f003:**
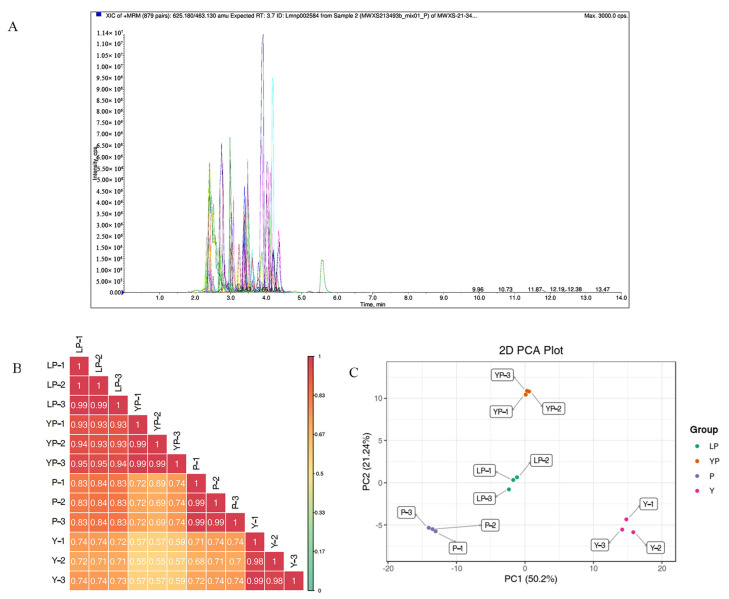
Overall qualitative and quantitative analyses of metabolomics data. (**A**) TIC overlap map of QC samples by MS detection. Abscissa represents the retention time (min) of metabolite detection. Ordinate represents the intensity of ion current (cps: count per second). (**B**) Pearson’s correlation coefficients among Y, LP, YP, and P. (**C**) PCA of Y, LP, YP, and P. *X*-axis represents the first principal component. *Y*-axis represents the second principal component.

**Figure 4 ijms-24-06472-f004:**
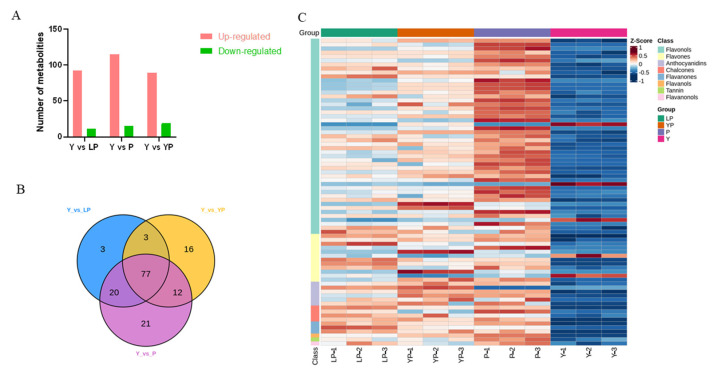
Analysis of differential metabolites. (**A**) The upregulated and downregulated differential metabolites in different petal groups. (**B**) Venn diagram of upregulated differential metabolites in different petal groups. (**C**) Heatmap of the differential metabolites. Red bars indicate highly accumulated metabolites, and blue indicates less accumulated metabolites.

**Figure 5 ijms-24-06472-f005:**
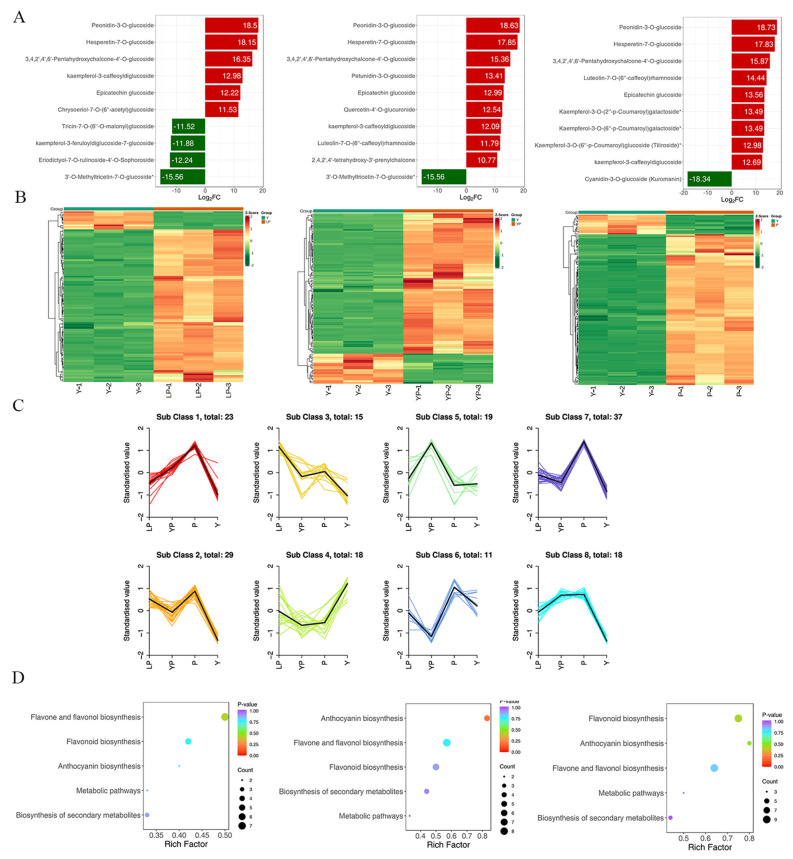
(**A**) Differential metabolites bar graph. Horizontal coordinates are log2FC of differential metabolites and vertical coordinates are differential metabolites. Red represents upregulated differential metabolites, and green represents downregulated differential metabolites. From left to right, Y vs. LP, Y vs. YP, Y vs. P. (**B**) Differential metabolites clustering heat map. From left to right, Y vs. LP, Y vs. YP, Y vs. P. (**C**) K means analysis. (**D**) KEGG analysis of differential metabolites. The horizontal coordinate indicates the Rich factor of each pathway, the vertical coordinate is the name of the pathway, and the color of the dot is the *p*-value, the redder it is, the more significant the analysis. The size of the dots represents the number of differential metabolites. From left to right, Y vs. LP, Y vs. YP, Y vs. P.

**Figure 6 ijms-24-06472-f006:**
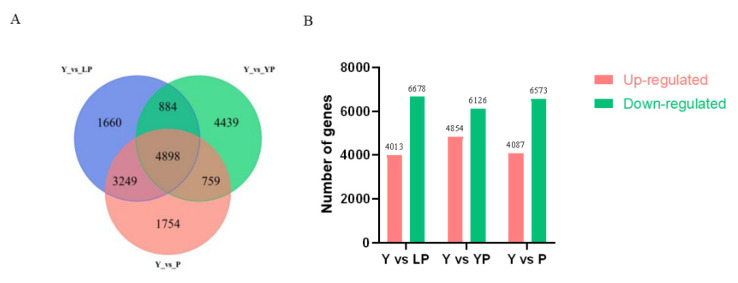
The DEGs between differently colored petals. (**A**) The upregulated and downregulated DEGs in different petal color groups. (**B**) Venn diagram of the upregulated DEGs between different petals color groups. Red bars indicate highly expressed genes, and green indicates less expressed genes.

**Figure 7 ijms-24-06472-f007:**
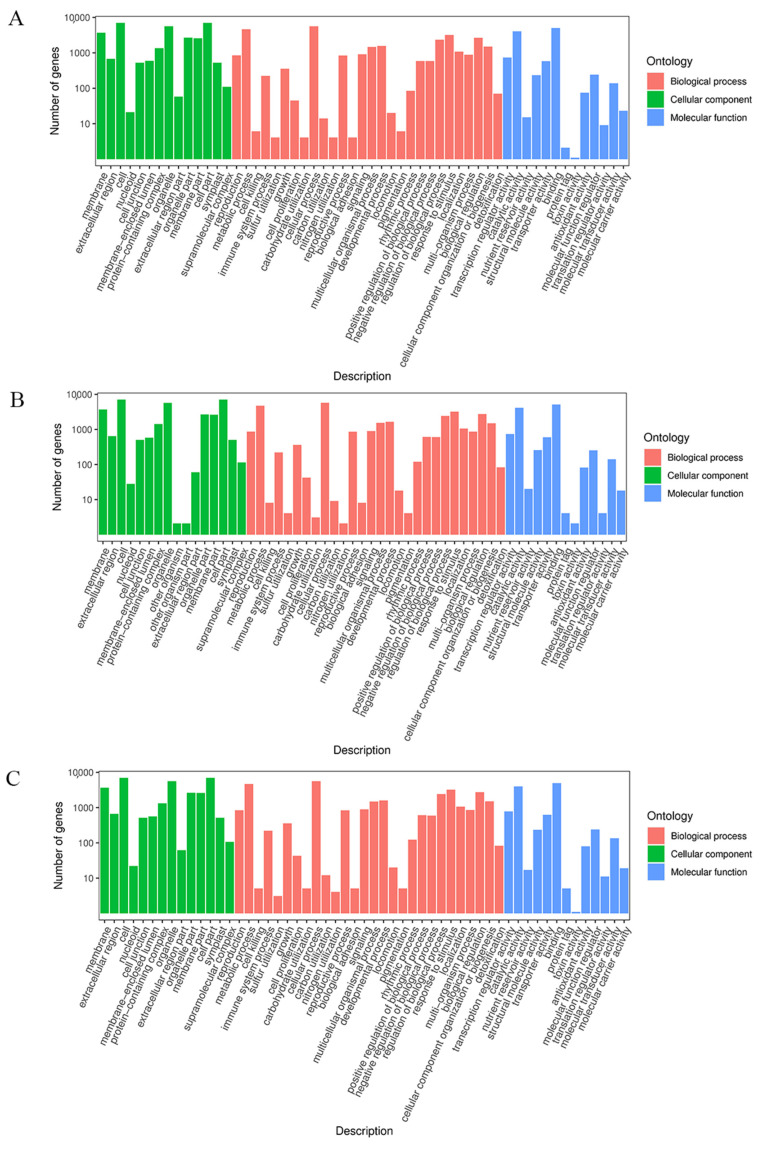
GO analysis of the DEGs. (**A**) GO functional classification of DEGs between Y and LP. (**B**) GO functional classification of DEGs between Y and YP. (**C**) GO functional classification of DEGs between Y and P. Red arrows indicate the DEGs associated with metabolic processes.

**Figure 8 ijms-24-06472-f008:**
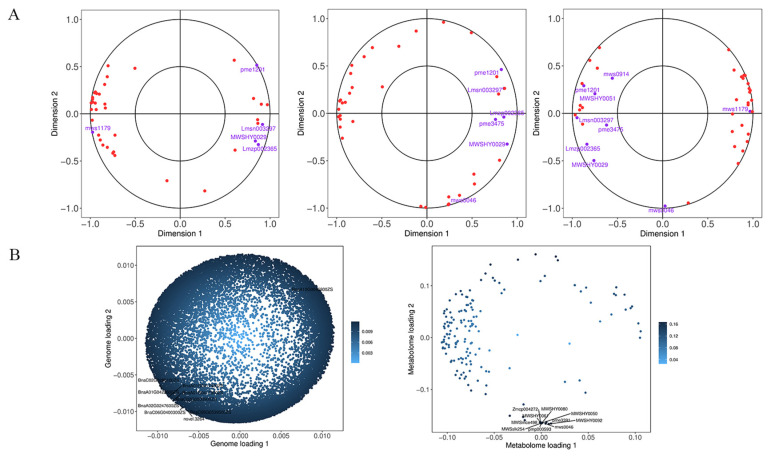
(**A**) Canonical correlation analysis (CCA). From left to right, Y vs. LP, Y vs. YP, Y vs. P. (**B**) Left, transcriptome loading plot; right, metabolome loading plot.

**Figure 9 ijms-24-06472-f009:**
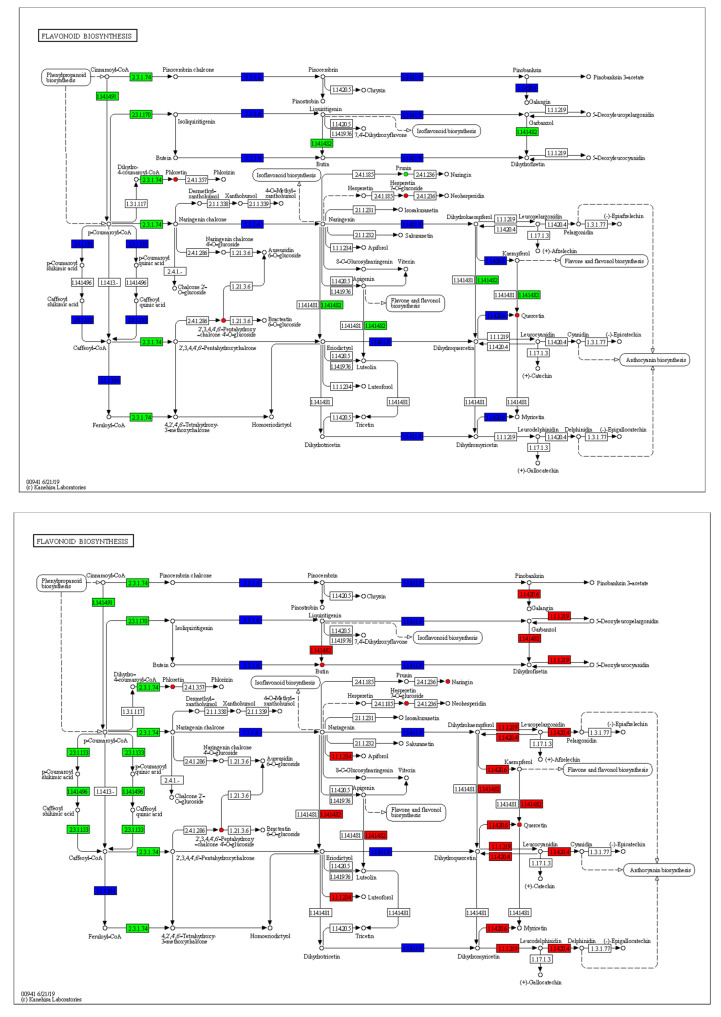
KEGG pathway analysis of flavonoid biosynthesis pathway. From top to bottom, Y vs. LP, Y vs. YP, Y vs. P.

**Figure 10 ijms-24-06472-f010:**
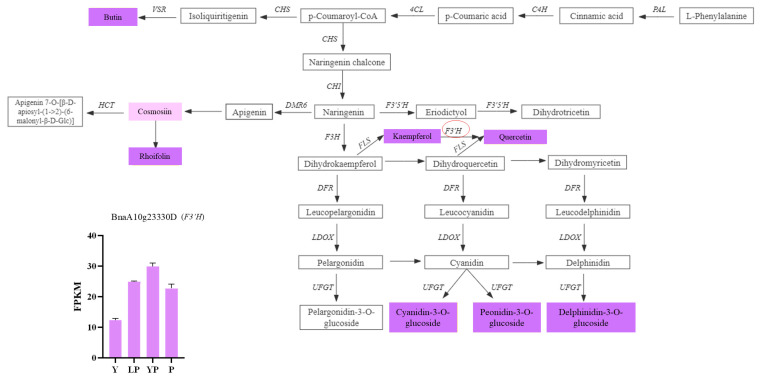
The flavonoid biosynthetic pathway. Heat maps of the DEGs between LP, YP, P, and Y. The light purple and purple boxes indicate significantly upregulated/downregulated metabolites in the flavonoid metabolic pathway. The *F3’H* gene (*BnaA10g23330D*) circled in red is the key differential gene in the flavonoid pathway that regulates purple flower traits.

**Figure 11 ijms-24-06472-f011:**
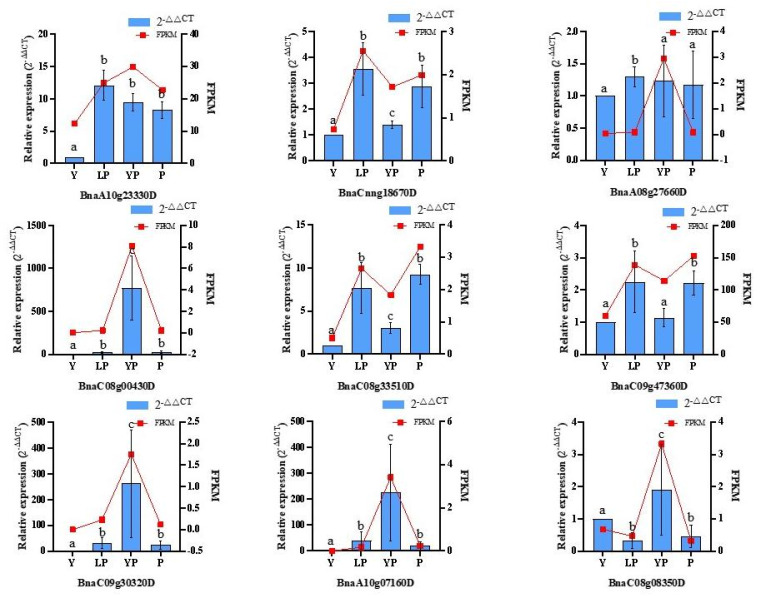
qRT-PCR analysis of selected DEGs in the rapeseed petal subtypes, identified by transcriptome analysis. Note: Y represents a yellow petal line; LP represents a light purple petal line; YP represents a yellow/purple petal line; P represents a purple petal line. *BnaA10g23330D*: *BnF3ʹH*; *BnaCnng18670D: BnVSR*; *BnaA08g27660D* and *BnaC08g00430D*: *BnHCT*; *BnaC08g33510D*: *BnFLS*; *BnaC09g47360D: BnF3H*; *BnaC08g08350D*, *BnaC09g30320D* and *BnaA10g07160D*: *BnUFGT*. Fold Change: Take Y as the reference point. Internal reference gene: *β*-actin. Data are means ± SD obtained from three biological replicates. Different letters above the error bars indicate significant differences at *p* < 0.05, according to Tukey’s test.

## Data Availability

The RNA-seq data of this study have been deposited in the National Center for Biotechnology Information (NCBI) database (accession number PRJNA942610, accessed on 21 February 2023).

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
