# Peer review of "Flavonoid Synthesis-Related Genes Determine the Color of Flower Petals in *Brassica napus* L."

_ijms, 2023, doi:10.3390/ijms24076472_

Round 1

Reviewer 1 Report (Previous Reviewer 1)

I thank the authors for their attempt to revise the manuscript. The authors added new graphical representations of their results, and one of my major issues has been addressed by the included PCAs for metabolomics. The same should be done for the reported NGS analyses. However, I am disappointed that most of my other issues have not been addressed.

× Missing statistical evaluation (Figures) - statistics in figures are still inadequate. For example, Figure 1 requires multivariant statistics. Its present form is most likely based on a poorly done Student's t-test (the meaning of asterisks is not indicated, and the methodology for statistics has not been improved) - the comparison with LP is missing. The number of biological/technical replicates is not indicated.  

× Missing the comparison of NGS and qPCR

The response to review indicates that this has been modified, but Figure 10 does not include correct statistical evaluation and does not indicate the observed regulation in the NGS. 

× Missing description of statistics in methods, missing explanation of the employed tests in tables (as indicated above, this has not been corrected)

× Missing more in-depth analysis of the NGS data. I believe that the authors should consider changing the approach as real data analysis does not seem to be their priority. Consider presenting NGS as a tool for the analysis of all known genes involved in flavonoid biosynthesis and removing comparisons that are not utilized in your study.  

- The authors have not modified this part, and the abstract highlights a very high number of identified DEGs that are not discussed in the manuscript or available in supplementary data. 

× Identified 'enrichment' in the flavonoid pathway - not a real enrichment for the targeted analysis of flavonoids

×Please, explain the difference between Figure 2A and Figure 3C/metabolomic data. I realized that the anthocyanin content was not measured in the LC-MS analysis, but it is still surprising that the anthocyanidin content seems to be much higher in LP than in P (YP>LP>P>Y) samples. I find it strange that this is not discussed in the manuscript.

×Language quality has not improved, and there are new mistakes:

  • 'white' is not a pigment, and its content does not exist
  • metabolites are not 'expressed'
  • OPLS does not contain 'principal components' - these are predictive and orthogonal components, and presented data do not provide indicated information about the model reliability

× The description of figures is still inadequate, newly introduced figures are of relatively low quality, and the supplementary figures are illegible. For example, Figure 5A - legend does not explain what is being represented by different comparisons. Why there are three OPLS models for the analysis? What was its purpose if the output is limited to pair-wise comparisons?

× Fold Change is listed in many comparisons, but it is not clear what is the reference point. 

× The description of methods has been improved, but this section still requires significant improvement, and some common mistakes were introduced in the revision (rpm value is meaningless without the corresponding parameters of the rotor - indicate RCF instead).

Author Response

Thank you for your letter and for the reviewers’ comments concerning our manuscript entitled “Flavonoid synthesis-related genes determine the colour of flower petals in Brassica napus L ” (ID: ijms-2167423). Those comments are all valuable and very helpful for revising and improving our paper, as well as the important guiding significance to our researches. We have studied comments carefully and have made correction which we hope meet with approval. The main corrections in the paper and the responds to the reviewer’s comments are as flowing:

Responds to the reviewer’s comments:

Reviewer #1:

  1. Response to comment: Moderate English changes required.

Response: As suggested by the reviewers, we have made appropriate changes to the English in the article.

  1. Response to comment: The introduction, references, experimental design, methods, results and conclusions need to be modified.

Response: As reviewer suggested that we have modified the above parts.

  1. Response to comment: Missing statistical evaluation (Figures).

Response: It is really true as reviewer suggested that we re-evaluated the statistics of Figures 2.We have improved methodology for statistics and added the meaning of letters. We added the number of biological/technical replicates.

  1. Response to comment: Missing the comparison of NGS and qPCR.

Response: It is really true as reviewer suggested that we have added correct statistical evaluation of figure 10 and indicated the observed regulation in the NGS.

  1. Response to comment: Missing description of statistics in methods, missing explanation of the employed tests in tables.

Response: It is really true as reviewer suggested that we added description of statistics in methods and we have made changes to supplementary table1.

  1. Response to comment: Missing more in-depth analysis of the NGS data.

Response: It is really true as reviewer suggested that we found the differential metabolites and regulated genes from the KEGG map of the flavonoid pathway enriched by the metabolome, and verified them with the KEGG map of the flavonoid pathway of the transcriptome, and supplemented the KEGG flavonoid pathway map of the two groups for joint analysis in Figure 7.

  1. Response to comment: Identified 'enrichment' in the flavonoid pathway.  

Response: It is really true as reviewer suggested that the targeted analysis of flavonoids is anthocyanidin pathway of flavonoid pathway. Flavonoids include anthocyanins.

  1. Response to comment: Please, explain the difference between Figure 2A and Figure 3C/metabolomic data.

Response: As suggested by the reviewers, we have retested the anthocyanin content of the petals of four flower colors. The results show that the content of anthocyanins in LP, YP and P petals is significantly higher than that in Y petals.

  1. Response to comment: Language quality has not improved, and there are new mistakes:

Response: As reviewer suggested that we revised the mistakes in the manuscript.

  1. Response to comment: The description of figures is still inadequate 

Response: It is really true as reviewer suggested that we put the original image into the compressed package and replaced the image in the manuscript.

  1. Response to comment: Fold Change is listed in many comparisons, but it is not clear what is the reference point.

Response: It is really true as reviewer suggested that we added the reference point is Y.

  1. Response to comment:The description of methods has been improved.

Response: It is really true as reviewer suggested that we have improved the description of methods.

We tried our best to improve the manuscript and made some changes in the manuscript. These changes will not influence the content and framework of the paper. We appreciate for Editors/Reviewers’ warm work earnestly, and hope that the correction will meet with approval.

Thank you very much for your suggestion, which has brought us great help.

Once again, thank you very much for your comments and suggestions.

Reviewer 2 Report (Previous Reviewer 2)

The novelty of the manuscript is quite low. A deep comparison with published papers recently is needed.

Author Response

Thank you for your letter and for the reviewers’ comments concerning our manuscript entitled “Flavonoid synthesis-related genes determine the colour of flower petals in Brassica napus L ” (ID: ijms-2167423). Those comments are all valuable and very helpful for revising and improving our paper, as well as the important guiding significance to our researches. We have studied comments carefully and have made correction which we hope meet with approval. The main corrections in the paper and the responds to the reviewer’s comments are as flowing:

Reviewer #2:

  1. Response to comment: English language and style are fine/minor spell check required.

Response: As suggested by the reviewers, we have made appropriate changes to the English in the article..

  1. Response to comment: The introduction, references, experimental design, methods, results and conclusions need to be modified.

Response: As reviewer suggested that we have modified the above parts.

  1. Response to comment: The novelty of the manuscript is quite low. A deep comparison with published papers recently is needed.

Response: It is really true as reviewer suggested that we have improved the discussion. The innovation of the manuscript is to select three kinds of Brassica napus and Brassica napus with different purple shades for transcriptome and flavonoid metabolome analysis. The material is relatively novel. Compared with carotenoid biosynthesis pathway, flavonoid biosynthesis pathway is selected to supplement previous studies and provide reference for molecular mechanism of purple rape.

We tried our best to improve the manuscript and made some changes in the manuscript. These changes will not influence the content and framework of the paper. We appreciate for Editors/Reviewers’ warm work earnestly, and hope that the correction will meet with approval.

Thank you very much for your suggestion, which has brought us great help.

Once again, thank you very much for your comments and suggestions.

Reviewer 3 Report (New Reviewer)

In my opinion Authors provided interesting results related to gene  expression and metabolite concentration that differ between four inbred lines of rapeseed that have different coloration.

Study is well planned and performed. Results suport conclusions. The material and methods section should be much better described. Following revisions should be included before the publication.

Section 3.1

How the inbred lines were obtained and characterized?

Section 3.2.1

How carotenoid and anthocyanin were extracted: describe if the wet or dry plant material was extracted, provide duration and temperature of extraction.

Section 3.3.1

Provide volume and concentration of libraries. Provide number of PCR reaction steps.

Citation provided in line 309 does not exist in References.

Section 3.3.2

Software used to trimming, trimming parameters, thresholds for data exclusion.

Source and version of the genome reference.

Quantification of gene expression; source and version of the software.

Normalisation method applied, package and version used.

Citation provided in line 317 does not exist in References.

Section 3.3.3.

Provide amount of RNA/cDNA per analysed sample.

Provide details of RT and PCR reaction.

Provide name of reference gene.

Was the stability of reference gene confirmed in earlier research or by analysis using specialized software as BestKeeper or related?

 Line 21 Rewrite sequence:

These genes may play a key role in regulating rape flower color, which is the next research object.

Line 217; Do not capitalize Genes.

Fig. 9; discriminate the up and down regulated metabolites by different colors. Try to mark metabolites responsible for purple plant color also by the purple color.

Line 87,289- correct spacing.

Line 329- provide citation of 2-ΔΔCT metod.

Fig. 10 provide information if differences were statistically significant.

Author Response

Thank you for your letter and for the reviewers’ comments concerning our manuscript entitled “Flavonoid synthesis-related genes determine the colour of flower petals in Brassica napus L ” (ID: ijms-2167423). Those comments are all valuable and very helpful for revising and improving our paper, as well as the important guiding significance to our researches. We have studied comments carefully and have made correction which we hope meet with approval. The main corrections in the paper and the responds to the reviewer’s comments are as flowing:

Reviewer #3:

  1. Response to comment: English language and style are fine/minor spell check required.

Response: As suggested by the reviewers, we have made appropriate changes to the English in the article.

  1. Response to comment: The introduction, references, experimental design, methods, results and conclusions need to be modified.

Response: As reviewer suggested that we have rewritten and modified the above parts.

  1. Response to comment: How the inbred lines were obtained and characterized?

Response: It is really true as reviewer suggested that the inbred line was obtained through population separation of line P01 and three generations of inbreeding. The identification method was color stability, consistent characters and no further separation.

  1. Response to comment: How carotenoid and anthocyanin were extracted: describe if the wet or dry plant material was extracted, provide duration and temperature of extraction.

Response: It is really true as reviewer suggested that carotenoids were extracted from wet samples in dark at room temperature for 3 hours, and anthocyanins were extracted from freeze-dried powder samples in dark at 4 degrees for 12 hours.

  1. Response to comment: Provide volume and concentration of libraries. Provide number of PCR reaction steps. Citation provided in line 309 does not exist in References.

Response: It is really true as reviewer suggested that we have added the contents that need to be supplemented to the annex and provided in line 309 in References.

  1. Response to comment: Software used to trimming, trimming parameters, thresholds for data exclusion. Source and version of the genome reference.Quantification of gene expression; source and version of the software.Normalisation method applied, package and version used. Citation provided in line 317 does not exist in References.

Response: It is really true as reviewer suggested that we have added the contents that need to be supplemented to the annex provided in line 317 in References

  1. Response to comment: Provide amount of RNA/cDNA per analysed sample. Provide details of RT and PCR reaction. Provide name of reference gene. Was the stability of reference gene confirmed in earlier research or by analysis using specialized software as BestKeeper or related?

Response: It is really true as Reviewer suggested that we provide amount of RNA/cDNA per analysed sample, details of RT and PCR reaction and name of reference gene. The stability of reference gene confirmed in earlier research

  1. Response to comment: Line 21 Rewrite sequence: These genes may play a key role in regulating rape flower color, which is the next research object.Line 217; Do not capitalize Genes.Fig. 9; discriminate the up and down regulated metabolites by different colors. Try to mark metabolites responsible for purple plant color also by the purple color. Line 87,289- correct spacing. Line 329- provide citation of 2-ΔΔCT metod. Fig. 10 provide information if differences were statistically significant.

Response: As suggested by the reviewers, we have corrected the above problems.

We tried our best to improve the manuscript and made some changes in the manuscript. These changes will not influence the content and framework of the paper. We appreciate for Editors/Reviewers’ warm work earnestly, and hope that the correction will meet with approval.

Thank you very much for your suggestion, which has brought us great help.

Once again, thank you very much for your comments and suggestions.

Reviewer 4 Report (New Reviewer)

The article "Flavonoid synthesis-related genes determine the color of flower petals in Brassica napus L." is devoted to the analysis of the genetic control of petal color in rapeseed (Brassica napus). The team of scientists used the entire set of research tools needed: biochemical analysis, RNAseq, qPCR and statistical analysis.
There are no complaints about the work either in terms of relevance and novelty. In my opinion, the presented study has both independent significance and can serve as a good example for similar work on other organisms.

However, there are minor editorial flaws:
line 162: Differentially expressed metabolite -- I'm a little confused by the wording, is this a well-established term
line 387: thirty-three, in my opinion, it is better to replace with numbers
Also, in the presented manuscript, the drawings have insufficient resolution.

Author Response

Thank you for your letter and for the reviewers’ comments concerning our manuscript entitled “Flavonoid synthesis-related genes determine the colour of flower petals in Brassica napus L ” (ID: ijms-2167423). Those comments are all valuable and very helpful for revising and improving our paper, as well as the important guiding significance to our researches. We have studied comments carefully and have made correction which we hope meet with approval. The main corrections in the paper and the responds to the reviewer’s comments are as flowing:

Reviewer #4:

  1. Response to comment: line 162: Differentially expressed metabolite -- I'm a little confused by the wording, is this a well-established term.

Response: It is really true as reviewer suggested that we have changed to differential metabolites.

  1. Response to comment: line 387: thirty-three, in my opinion, it is better to replace with numbers.

Response: As reviewer suggested that we have changed to 33.

  1. Response to comment: Also, in the presented manuscript, the drawings have insufficient resolution.

Response: It is really true as reviewer suggested that we put the original image into the compressed package and replaced the image in the manuscript.

We tried our best to improve the manuscript and made some changes in the manuscript. These changes will not influence the content and framework of the paper. We appreciate for Editors/Reviewers’ warm work earnestly, and hope that the correction will meet with approval.

Thank you very much for your suggestion, which has brought us great help.

Once again, thank you very much for your comments and suggestions

Round 2

Reviewer 1 Report (Previous Reviewer 1)

I am confused with the response to the review and the indicated modifications in the manuscript. It could be possible that the wrong version was uploaded. As it stands, there is some improvement, but not sufficient.

Issues not addressed:

Figure 2 - description has not been modified to address my comments. 

Figure 3 - Figure resolution is too low and fonts are illegible

Figure 5 - low resolution, illegible fonts

Figure 8 - low resolution, illegible fonts

Language still requires editing, the authors should consider involving a native speaker.

L149 Enrichment in metabolome - again, this is a targeted analysis of flavonoids. You can not comment on "enrichment" of these metabolites or the corresponding metabolic pathways.  

× Missing the comparison of NGS and qPCR

The response to review indicates that this has been modified, but Figure 10 does not include correct statistical evaluation and does not indicate the observed regulation in the NGS. 

-not addressed

× Missing more in-depth analysis of the NGS data. I believe that the authors should consider changing the approach as real data analysis does not seem to be their priority. Consider presenting NGS as a tool for the analysis of all known genes involved in flavonoid biosynthesis and removing comparisons that are not utilized in your study.  

- The authors have not modified this part, and the abstract highlights a very high number of identified DEGs that are not discussed in the manuscript or available in supplementary data. 

-not addressed

×Please, explain the difference between Figure 2A and Figure 3C/metabolomic data...

response: As suggested by the reviewers, we have retested the anthocyanin content of the petals of four flower colors. The results show that the content of anthocyanins in LP, YP and P petals is significantly higher than that in Y petals.

I did not question the low amount found in the Y samples. My issue was with a much higher content of anthocyanidins in LP than in P that does not match anthocyanin accumulation.

Anthocyanins are the glycosylated form of anthocyanidins. The surprising difference between anthocyanin and anthocyanidin content should be at least discussed. 

Statistics

p-value is indicated in Materials and Methods (L335), but the employed statistical test is not. 

Data Availability

This is a descriptive study reporting thousands of DEGs and DAMs. The statement should indicate that these data are in the supplementary files, and the NGS data should be accessible via a public repository.

Author Response

Dear Editors and Reviewers:

Thank you for your letter and for the reviewers’ comments concerning our manuscript entitled “Flavonoid synthesis-related genes determine the colour of flower petals in Brassica napus L ” (ID: ijms-2167423). Those comments are all valuable and very helpful for revising and improving our paper, as well as the important guiding significance to our researches. We have studied comments carefully and have made correction which we hope meet with approval. The main corrections in the paper and the responds to the reviewer’s comments are as flowing:

Responds to the reviewer’s comments:

Reviewer #1:

  1. Response to comment: Moderate English changes required. Language still requires editing, the authors should consider involving a native speaker.

Response: As suggested by the reviewers, we asked the company MogoEdit to edit the manuscript in English.

  1. Response to comment: Figure 2 - description has not been modified to address my comments. Figure 3 - Figure resolution is too low and fonts are illegible. Figure 5and Figure 8- low resolution, illegible fonts.

Response: As reviewer suggested that we have modified the following image and put the original image into the compressed package for you to view.

  1. Response to comment: L149 Enrichment in metabolome.

Response: It is really true as reviewer suggested that we modified this part in Section 2.3 (L243-248).

  1. Response to comment: Missing the comparison of NGS and qPCR.

Response: It is really true as reviewer suggested that we have added correct statistical evaluation of figure 11 and indicated the observed regulation in the NGS (Section 2.8 L427-428).

  1. Response to comment: Missing description of statistics in methods, missing explanation of the employed tests in tables.

Response: It is really true as reviewer suggested that we added description of statistics in methods and we have made changes to ALL DEGS supplementary table1.

  1. Response to comment: Missing more in-depth analysis of the NGS data.

Response: It is really true as reviewer suggested that we found the differential metabolites and regulated genes from the KEGG map of the flavonoid pathway enriched by the metabolome, and verified them with the KEGG map of the flavonoid pathway of the transcriptome, and supplemented the KEGG flavonoid pathway map of the two groups for joint analysis in Figure 9.

  1. 7. Response to comment: Please, explain the difference between Figure 2A and Figure 3C/metabolomic data.  

Response: It is really true as reviewer suggested that we explained the difference between Figure 2A and Figure 3C (now Figure 4C)/metabolomic data in Discussion (L823-828).

  1. 8. Response to comment: Statistics.

Response: As suggested by the reviewers, we have added the employed statistical test in Section 3.3.4 (L636-639).

  1. 9. Response to comment: Data Availability.

Response: As reviewer suggested that we modified this part in Section 2.3 (L243-248).

We tried our best to improve the manuscript and made some changes in the manuscript. These changes will not influence the content and framework of the paper. We appreciate for Editors/Reviewers’ warm work earnestly, and hope that the correction will meet with approval.

Thank you very much for your suggestion, which has brought us great help.

Once again, thank you very much for your comments and suggestions.

Reviewer 2 Report (Previous Reviewer 2)

Novelty is quiet low.

Author Response

Dear Editors and Reviewers:

Thank you for your letter and for the reviewers’ comments concerning our manuscript entitled “Flavonoid synthesis-related genes determine the colour of flower petals in Brassica napus L ” (ID: ijms-2167423). Those comments are all valuable and very helpful for revising and improving our paper, as well as the important guiding significance to our researches. We have studied comments carefully and have made correction which we hope meet with approval. The main corrections in the paper and the responds to the reviewer’s comments are as flowing:

Reviewer #2:

  1. Response to comment: English language and style are fine/minor spell check required.

Response: As suggested by the reviewers, we asked the company MogoEdit to edit the manuscript in English.

  1. Response to comment: The introduction, references, experimental design, methods, results and conclusions need to be modified.

Response: As reviewer suggested that we have modified the above parts.

  1. Response to comment: Novelty is quiet low..

Response: It is really true as reviewer suggested that we have improved the discussion.  We selected three kinds of Brassica napus with different degrees of purple to compare the petals with those of yellow rape, and they have the middle color of yellow and purple, which is different from other people's research. Our study found the HCT gene that may regulate the yellow and purple petal character. And found the metabolite butin and phloetin which have both medicinal value and are related to flower color. Our research shows that the content of peonidin-3-O-glucoside, delphinidin-3-O-glucoside is closely related to the purple flower character, which is consistent with previous research. Our research shows that F3'H gene regulates purple flower traits, which is consistent with previous studies

We tried our best to improve the manuscript and made some changes in the manuscript. These changes will not influence the content and framework of the paper. We appreciate for Editors/Reviewers’ warm work earnestly, and hope that the correction will meet with approval.

Thank you very much for your suggestion, which has brought us great help.

Once again, thank you very much for your comments and suggestions.

Reviewer 3 Report (New Reviewer)

Authors significantly improved manuscript. Minor corrections should be done:

1. Provide volume and concentration of cDNA library.

2. Provide details of RT-PCR cycling reaction, temperatures and times of denaturation, annealing and extension steps etc.

3. Provide amount of RNA/cDNA used in RT-PCR experiment.

Author Response

Dear Editors and Reviewers:

Thank you for your letter and for the reviewers’ comments concerning our manuscript entitled “Flavonoid synthesis-related genes determine the colour of flower petals in Brassica napus L ” (ID: ijms-2167423). Those comments are all valuable and very helpful for revising and improving our paper, as well as the important guiding significance to our researches. We have studied comments carefully and have made correction which we hope meet with approval. The main corrections in the paper and the responds to the reviewer’s comments are as flowing:

Reviewer #3:

  1. Response to comment: English language and style are fine/minor spell check required.

Response: As suggested by the reviewers, we asked the company MogoEdit to edit the manuscript in English.

  1. Response to comment: The introduction, references, experimental design, methods, results and conclusions need to be modified.

Response: As reviewer suggested that we have rewritten and modified the above parts.

  1. Response to comment: Provide volume and concentration of cDNA library.

Response: It is really true as reviewer suggested that we provided volume and concentration of cDNA library in Section 3.3.1 (L595-596).

  1. Response to comment: Provide details of RT-PCR cycling reaction, temperatures and times of denaturation, annealing and extension steps etc.

Response: It is really true as reviewer suggested that we provided details of RT-PCR cycling reaction, temperatures and times of denaturation, annealing and extension steps etc in Section 3.3.1 (L596-600).

  1. Response to comment: Provide amount of RNA/cDNA used in RT-PCR experiment..

Response: It is really true as reviewer suggested that we provided amount of RNA/cDNA used in RT-PCR experiment in Section 3.3.3 (L618).

We tried our best to improve the manuscript and made some changes in the manuscript. These changes will not influence the content and framework of the paper. We appreciate for Editors/Reviewers’ warm work earnestly, and hope that the correction will meet with approval.

Thank you very much for your suggestion, which has brought us great help.

Once again, thank you very much for your comments and suggestions.

Round 3

Reviewer 1 Report (Previous Reviewer 1)

  1. Response: As suggested by the reviewers, we asked the company MogoEdit to edit the manuscript in English.
    × The language of the manuscript has significantly improved, and this issue has been addressed.
  1. Response: As reviewer suggested that we have modified the following image and put the original image into the compressed package for you to view.
    ×  I don't believe that the compressed image was included. It is possible that this content was not forwarded to the reviewers (?) Nevertheless, the fonts are too small for the final figure and legibility will always be an issue unless that is solved. 
  1. Response: It is really true as reviewer suggested that we modified this part in Section 2.3 (L243-248).
    × It seems that my comments have not been clear enough. Enrichment analysis is based on comparing your list with that of all possible pathways. If your list is based on targeted analysis of flavonoids, that is not enrichment. You have only flavonoids in your metabolome. By combining metabolome and transcriptome data, the enrichment can not be discussed. Again, you are using only a targeted analysis. That description of your data is technically incorrect. This issue can be easily addressed by modifying the corresponding text and excluding comments about "enrichment" in metabolome data. 
  1. Response: It is really true as reviewer suggested that we have added correct statistical evaluation of figure 11 and indicated the observed regulation in the NGS (Section 2.8 L427-428).
    × Lines 427-428 are deleted in the manuscript and are not in Section 2.8. The text in that section seems to be misordered - L287-288 should be preceded by the description in L288-292. It seems that the red line in the figure corresponds to the expected profile. The abbreviation should be listed in the figure legend. Furthermore, there are definitely some differences that should be at least mentioned, and the statement about consistency should be modified (at best "mostly consistent") - the FPKM Profile in BnaA08g27660D does not match that of qRT-PCR, and Y - YP difference was not validated for BnaC09g47360D. I also noticed some errors in the reported posthoc analyses (BnaC08g08350D). 
  1. Response: It is really true as reviewer suggested that we added description of statistics in methods and we have made changes to ALL DEGS supplementary table1.
    × The provided statistics are incomplete, but the table includes raw data and it can be easily reanalyzed if needed. This issue has been addressed. 
  1. Response: It is really true as reviewer suggested that we found the differential metabolites and regulated genes from the KEGG map of the flavonoid pathway enriched by the metabolome, and verified them with the KEGG map of the flavonoid pathway of the transcriptome, and supplemented the KEGG flavonoid pathway map of the two groups for joint analysis in Figure 9.
    × I am not satisfied with the revision, but this point is not critical for the manuscript acceptance. The only needed revision is in the abstract - modify that to indicate that a reader will not find an in-depth analysis of 10,000 DEGs in this manuscript.  
  1. Response: It is really true as reviewer suggested that we explained the difference between Figure 2A and Figure 3C (now Figure 4C)/metabolomic data in Discussion (L823-828).
    × There is no line 823 in the revised manuscript, with the last line being L795 (references). I searched the text for the Figure 2A reference, but all were deleted. I believe that this issue is not addressed. 
  1. Response: As suggested by the reviewers, we have added the employed statistical test in Section 3.3.4 (L636-639).
    × This issue was addressed. 
  1. Response: As reviewer suggested that we modified this part in Section 2.3 (L243-248).
    × The omics data should be deposited in a public repository. The absence of raw files is not recommended. However, most readers will be satisfied with the newly provided supplementary tables. I will leave this decision to the MDPI editors. 

The manuscript has been improved. It is far from perfect, but I believe that the remaining issues can be addressed with a minor revision.

Author Response

Dear Editors and Reviewers:

Thank you for your letter and for the reviewers’ comments concerning our manuscript entitled “Flavonoid synthesis-related genes determine the colour of flower petals in Brassica napus L ” (ID: ijms-2167423). Those comments are all valuable and very helpful for revising and improving our paper, as well as the important guiding significance to our researches. We have studied comments carefully and have made correction which we hope meet with approval. The main corrections in the paper and the responds to the reviewer’s comments are as flowing:

Responds to the reviewer’s comments:

Reviewer 1:

  1. Response to comment: English language and style are fine/minor spell check required.

Response: As suggested by the reviewer, we contracted MogoEdit company to edit the revised manuscript.

  1. Response to comment: Figure resolution is too low and fonts are illegible.

Response: It is really true as reviewer suggested that we adjusted fonts in figures and put the original image into the compressed package for comparison.

  1. Response to comment: L149 Enrichment in metabolome.

Response: It is really true as reviewer suggested that we modified the corresponding text and excluded comments about "enrichment" in metabolome data.

  1. Response to comment: Missing the comparison of NGS and qPCR.

Response: It is really true as reviewer suggested that we list the abbreviations in the legend and revised the contents of this section. The original text is as follows: To verify the differential expression of the genes identified by transcriptome analysis, we used qRT-PCR to analyze the relative expression levels of nine selected genes involved in flavonoid metabolism (HCT, VSR7, F3H, F3’H, FLS, and UFGT).The FPKM curve in BnaA08g27660D did not match the qRT-PCR, and the expression of BnaC09g47360D did not differ significantly between Y and YP. The relative expression of other genes was basically consistent with the transcriptome results, among which the F3’H gene was the key gene for regulating purple flower traits. The relative ex-pression of BnaA10g23330D gene in LP, YP and P was significantly higher than that in Y. Transcriptome results show showed  show that the relative expression of gene BnaA10g23330D in Y, LP, YP and P was 1, 11.93, 9.38 and 8.17, respectively.(Section2.8  L234-244)

  1. Response to comment: The only needed revision is in the abstract - modify that to indicate that a reader will not find an in-depth analysis of 10,000 DEGs in this manuscript. .

Response: It is really true as reviewer suggested that we modified the abstract. The original text is as follows: A total of 20511 differentially expressed genes (DEGs) between P, LP, YP, versus Y plant petals were detected. This study focused on the co-regulation of 4898 differential genes in the three comparison groups.(Abstract L15-17)

  1. Response to comment: Please, explain the difference between Figure 2A and Figure 3C/metabolomic data

Response: We are sorry for the change in the number of lines of the manuscript due to the revision. As suggested by the reviewer, we discussed the difference of anthocyanin content between the two figures in discussion. The original text is as follows: Anthocyanins are the glycosylated form of anthocyanidins. We found that the content of anthocyanins and anthocyanidins in the three purple petals was higher than that of yellow petals, but the accumulation of anthocyanins in the three purple petals differed from the accumulation of anthocyanidins. This may be attributed to the limited scope of the detection technology. At present, the detectable anthocyanin substances in the flavonoid metabolism group cannot cover all the anthocyanin categories in nature.(Discussion  L432-437)

  1. Response to comment: Data Availability.

Response: It is really true as reviewer suggested that we will refer to your suggestion to consider putting the omics data into the public database. We are happy to provide raw data from NGS. Because it takes some time to reside in a public database, the original data was not uploaded this time. If it needs to be stored in a public database, we will store it immediately.

We tried our best to improve the manuscript and made some changes in the manuscript. These changes will not influence the content and framework of the paper. We appreciate for Editors/Reviewers’ warm work earnestly, and hope that the correction will meet with approval.

Thank you very much for your suggestion, which has brought us great help.

Once again, thank you very much for your comments and suggestions.

This manuscript is a resubmission of an earlier submission. The following is a list of the peer review reports and author responses from that submission.

Round 1

Reviewer 1 Report

The manuscript by Shijun Li et al. presents the results of transcriptomic and metabolomic analysis of four different Brassica napus genotypes. The authors selected genotypes with contrasting content of pigments in petals and present candidate genes and metabolites responsible for the observed differences in the petal color.  The approach per se is not novel and it is surprising that the authors are not referencing recent studies published in this area (e.g., Yingjun Liu et al., 2020; 10.1111/tpj.14970) 

I believe that the content of the manuscript could be of interest, but it is not possible to evaluate that in its present version. The inadequate/missing statistics indicate a need for an extensive revision that is not feasible within the time given for major revisions. 

Major issues

1/ The authors claim that more than 200 differentially abundant metabolites and more than 4,000 DEGs were identified. These data are not supported by any of the enclosed Figures/Tables or Supplementary Tables. The whole manuscript is targeted at a preselected subset of genes and metabolites, and the identity of DEGs and identified metabolites is not disclosed. The authors must provide corresponding tables and show reproducibility and reliability of their data.

2/ Statistical evaluation of the results is missing

The experiment design requires an advanced statistical analysis that would identify/confirm metabolites and DEGs responsible for the observed pigmentation. However, the authors do not provide any statistical evaluation of the data. All figures must be supplemented with adequate statistical evaluation, including statistical tests and a description about the number of biological/technical replicates. Without that, the manuscript should  not be considered for publishing in a Q1 journal. 

Minor issues
1/ The manuscript would benefit from proofreading and editing; the style does not adhere to the recommended IJMS style. The order of chapters is confusing (2.1.2. - GO and KEGG analysis of DEG followed by a chapter describing DEG identification; Methods are not position correctly). Some sentences are too generalized to be correct (e. g., rapeseed is not used for honey, it is used for feeding bees).

2/ The manuscript contains many statements that are not supported by the presented data. For instance, the authors claim that qPCR data and NGS were consistent (L127-L128) - that must be shown and compared (at least in supplements). 

3/ The description in methods is insufficient and requires more details. 

4/ Figures require polishing and corrections.

For instance, Figure 5C:

Besides missing statistics, a R-genereated heatmap contains incorrect colorcoding (0 is not white) and indicates a high variability in replicates.

The description is missing - what is 'Z-score'?  

'Red indicates highly accumulated metabolites, and green indicates less accumulated metabolites.' - There are at least three shades of green in this figure. None of these represents abundance. 

5/ Abbreviations in the key words - what does UPLCI-MS/MS stands for?

Author Response

Dear Editors and Reviewers:

Thank you for your letter and for the reviewers’ comments concerning our manuscript entitled “Comparative analysis of three purple and one yellow-flowered rapeseed lines (Brassica napus L.) using UPLC-MS/MS and transcriptome analysis ” (ID:ijms-2030555). Those comments are all valuable and very helpful for revising and improving our paper, as well as the important guiding significance to our researches. We have studied comments carefully and have made correction which we hope meet with approval. Revised portion are marked in red in the paper. The main corrections in the paper and the responds to the reviewer’s comments are as flowing:

Responds to the reviewer’s comments:

Reviewer #1:

  1. Response to comment: The authors claim that more than 200 differentially abundant metabolites and more than 4,000 DEGs were identified. These data are not supported by any of the enclosed Figures/Tables or Supplementary Tables. The whole manuscript is targeted at a preselected subset of genes and metabolites, and the identity of DEGs and identified metabolites is not disclosed. The authors must provide corresponding tables and show reproducibility and reliability of their data.

Response: As Reviewer suggested that we added the identified differential genes and metabolites to the supplementary table.

  1. Response to comment: Statistical evaluation of the results is missing.

The experiment design requires an advanced statistical analysis that would identify/confirm metabolites and DEGs responsible for the observed pigmentation. However, the authors do not provide any statistical evaluation of the data. All figures must be supplemented with adequate statistical evaluation, including statistical tests and a description about the number of biological/technical replicates.

Response: We are very sorry for our negligence of statistical evaluation of the results. We conducted statistical evaluation for all figures.

  1. Response to comment: The manuscript would benefit from proofreading and editing; the style does not adhere to the recommended IJMS style.

Response: We have revised the structure of the article according to the Reviewer’s suggestion.

  1. Response to comment: The manuscript contains many statements that are not supported by the presented data. For instance, the authors claim that qPCR data and NGS were consistent (L127-L128) - that must be shown and compared (at least in supplements).

Response: It is really true as Reviewer suggested that we added NGS to the supplementary table.

  1. Response to comment: The description in methods is insufficient and requires more details. 

Response: We have revised the structure of the article according to the Reviewer’s suggestion.

  1. Response to comment: Figures require polishing and corrections.

Response: It is really true as Reviewer suggested that we added NGS to the supplementary table.

  1. Response to comment: Abbreviations in the key words - what does UPLCI-MS/MS stands for?

Response: We are very sorry for our negligence and spelling mistakes. UPLC-MS/MS stands for Ultra-performance liquid chromatography-tandem mass spectrometry.

  1. Response to comment: Figures require polishing and corrections.

Response: It is really true as Reviewer suggested that we added clearer pictures.

We tried our best to improve the manuscript and made some changes in the manuscript. These changes will not influence the content and framework of the paper. We appreciate for Editors/Reviewers’ warm work earnestly, and hope that the correction will meet with approval.

Special thanks to you for your good comments.

Once again, thank you very much for your comments and suggestions.

Reviewer 2 Report

Please reanalyze their data by combing MS and RNAseq data and reorganize the manuscript following similar work which published in 2022. If the authors focused on the petal color of Brassica napus, please show the specific color formation related genes and metabolites, not just state your data roughly.

1. Hao, Pengfei, et al. "BnaA03. ANS Identified by Metabolomics and RNA-seq Partly Played Irreplaceable Role in Pigmentation of Red Rapeseed (Brassica napus) Petal." Frontiers in Plant Science 13 (2022).

2. Ye, Shenhua, et al. "Genetic and multi-omics analyses reveal BnaA07. PAP2 In-184-317 as the key gene conferring anthocyanin-based color in Brassica napus flowers." Journal of Experimental Botany 73.19 (2022): 6630-6645.

Author Response

Dear Editors and Reviewers:

Thank you for your letter and for the reviewers’ comments concerning our manuscript entitled “Comparative analysis of three purple and one yellow-flowered rapeseed lines (Brassica napus L.) using UPLC-MS/MS and transcriptome analysis ” (ID:ijms-2030555). Those comments are all valuable and very helpful for revising and improving our paper, as well as the important guiding significance to our researches. We have studied comments carefully and have made correction which we hope meet with approval. Revised portion are marked in red in the paper. The main corrections in the paper and the responds to the reviewer’s comments are as flowing:

Responds to the reviewer’s comments:

Response to comment: Please reanalyze their data by combing MS and RNAseq data and reorganize the manuscript following similar work which published in 2022.

Response: It is really true as Reviewer suggested that we reanalyze their data and reorganize the manuscript.We referred to the article you recommended。

Special thanks to you for your comments.

We tried our best to improve the manuscript and made some changes in the manuscript. These changes will not influence the content and framework of the paper. We appreciate for Editors/Reviewers’ warm work earnestly, and hope that the correction will meet with approval.

Once again, thank you very much for your comments and suggestions.

Round 2

Reviewer 1 Report

Dear Authors,

I appreciate the efforts that were put into the revision, and I am glad that at least some of my major concerns have been addressed. 

Issues not addressed:

× Incomplete descriptions of figures

× Missing statistical evaluation (Figures)

× Missing the comparison of NGS and qPCR

× Missing description of statistics in methods, missing explanation of the employed tests in tables

× Missing evaluation of reproducibility 

× Missing more in-depth analysis of the NGS data. I believe that the authors should consider changing the approach as real data analysis does not seem to be their priority. Consider presenting NGS as a tool for the analysis of all known genes involved in flavonoid biosynthesis and removing comparisons that are not utilized in your study.  Table 1 - There are more than 10,000 DEGs and five "main pathways" represent less than 200 DEGs. That is not plausible. 

Additional issues based on newly incorporated text and supplementary files:

×The differential gene obtained from transcriptome analysis and the differential metabolite obtained from metabolome analysis were annotated with KEGG pathway, and the two omics data were mutually verified to determine that the metabolic pathway of key changes was flavonoid synthesis pathway and anthocyanin synthesis pathway. KEGG 159 functional annotation analysis showed that nine DEGs were enriched in flavonoid metabolism pathways (Supplementary Table). These genes affect the synthesis of flavonol and anthocyanin, thus leading to different flower colors.

That comparison is meaningless. Your metabolome analysis is the targeted one. All your metabolites are flavonoids. How could you find enrichment in a different metabolic pathway?

× The referenced manuscript (Yang et al., 2022) does not contain any information about data processing and QC runs that are included in the table. Please, provide a full description of the employed method, normalization, and data processing.

× Please, clarify the reason for excluding more than 50% of targeted metabolites. It seems that the analysis is targeted only at the subset of metabolites that are found in all genotypes. That does not make much sense, given the objective of identifying metabolites responsible for the observed differences.

× Please, explain the difference between Figure 2A and Figure 3C. The flavonoid content in Y is matching the expected trend, but LP is not. 

× The manuscript still requires proofreading and editing (e.g., "GO analysis of the DEGs between xyz."; "DEGs involved in the flavonoid and flavonoid synthesis"...)

Taken together, I believe that the manuscript was improved, but the improvement is not sufficient. I am optimistic that the authors will be able to address these in the revision. However, I will recommend the manuscript rejection if these issues are not addressed. 

Reviewer 2 Report

No any improvements on the manuscript contents.